# REINFORCEMENT LEARNING WITHOUT GROUND-TRUTH STATE

## ABSTRACT

To perform robot manipulation tasks, a low-dimensional state of the environment typically needs to be estimated. However, designing a state estimator can sometimes be difficult, especially in environments with deformable objects. An alternative is to learn an end-to-end policy that maps directly from high-dimensional sensor inputs to actions. However, if this policy is trained with reinforcement learning, then without a state estimator, it is hard to specify a reward function based on high-dimensional observations. To meet this challenge, we propose a simple indicator reward function for goal-conditioned reinforcement learning: we only give a positive reward when the robot's observation exactly matches a target goal observation. We show that by relabeling the original goal with the achieved goal to obtain positive rewards (Andrychowicz et al., 2017), we can learn with the indicator reward function even in continuous state spaces. We propose two methods to further speed up convergence with indicator rewards: reward balancing and reward filtering. We show comparable performance between our method and an oracle which uses the ground-truth state for computing rewards. We show that our method can perform complex tasks in continuous state spaces such as rope manipulation from RGB-D images, without knowledge of the ground-truth state.

## 1 INTRODUCTION

To perform robot manipulation tasks, a low dimensional state of the environment typically needs to be estimated. In reinforcement learning, this state is also used to compute the reward function. However, designing a state estimator can be difficult, especially in environments with deformable objects, as shown in Figure 1. An alternative is to learn an end-to-end policy that maps directly from high-dimensional sensor input to actions. However, without a state estimator, it is hard to specify a reward function based on high-dimensional observations.

Past efforts to use reinforcement learning for robotics have avoided this issue in a number of ways. One common approach is to use extra sensors to determine the state of the environment during training, even if such sensors are not available at test time. Examples of this include using another robot arm to hold all relevant objects (Levine et al., 2016), placing an IMU sensor (Gu

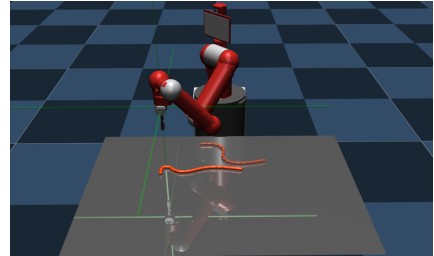

Figure 1: An illustration of the rope pushing task. The Sawyer robot is given an image of the current configuration of the rope and an image of the goal configuration (illustrated as the translucent rope) and the task is to push the rope to the goal configuration.

et al., 2017; Yahya et al., 2017), motion capture markers on such objects (Kormushev et al., 2010), or ensuring that all relevant objects are placed on scales (Schenck & Fox, 2016).

However, such instrumentation is not always easy to set up for each task. This is especially true for deformable object manipulation, such as rope or cloth manipulation, in which every part of the object must be instrumented in order to measure the full state of the entire object. Attaching such sensors to food or granular material would present additional difficulties.

We present an alternative approach for goal-conditioned reinforcement learning for specifying rewards using raw (e.g. high-dimensional and continuous) observations without requiring explicit state estimation or access to the ground-truth state of the environment. We achieve this using a

simple indicator reward function, which only gives a positive reward when the robot's observation exactly matches a target goal observation. Naturally, in continuous state spaces, we do not expect any two observed states to be identical. Surprisingly, we show that we can learn with such an indicator reward, even in continuous state spaces, if we use goal relabeling (Kaelbling, 1993; Andrychowicz et al., 2017), which relabels the original goal with the achieved observation such that a positive reward is given. As the indicator rewards produce extreme sparse positive rewards, we further introduce reward balancing to balance the positive and negative rewards, as well as reward filtering to filter out uncertain rewards.

We show theoretically that the indicator reward results in a policy with bounded suboptimality compared to the ground-truth reward. We also empirically show comparable performance between our method and an oracle which uses the ground-truth state for computing rewards, even though our method only operates on raw observations and does not have access to the ground-truth state. We demonstrate that an indicator reward can be used to teach a robot complex tasks such as rope manipulation from RGB-D images, without knowledge of the ground-truth state during training. Videos of our method can be found at `https://sites.google.com/view/image-rl`.

## 2 RELATED WORK

### 2.1 OBTAINING GROUND-TRUTH STATE FOR TRAINING

**Adding sensors:** To obtain ground-truth states for calculating rewards, one approach is to perform state estimation. However, such an approach can be noisy and challenging to implement, especially for the deformable objects that we study in this work. Another approach is to add extra sensors during training to accurately record the state. For example, in past work, one robot arm (covered with a cloth at training time) is used to rigidly hold and move an object, while another robot arm learns to manipulate the object (Levine et al., 2016). In such a case, the object position can be inferred directly from the position of the robot gripper that is holding it. In other work on teaching a robot to open a door, an IMU sensor is placed on the door handle to determine the rotation angle of the handle and whether or not the door has been opened (Gu et al., 2017; Yahya et al., 2017). One can also ensure that all relevant objects for a task are placed on scales (Schenck & Fox, 2016) or affixed with motion capture markers to obtain a precise estimate of their position (Kormushev et al., 2010). However, such instrumentation is challenging for deformable objects, granular material, food, or other settings. Further, such instrumentation is costly and time-consuming to setup; hence most of these previous approaches assume that such instrumentation is only available at training time and these methods do not allow further fine-tuning of the policy after deployment.

**Training in simulation:** Another common approach is to train the policy entirely in simulation in which ground-truth state can be obtained from the simulator (Fang et al., 2018; Andrychowicz et al., 2018b; Zhu et al., 2018; Sadeghi & Levine, 2016; Pinto et al., 2018). Many approaches have been explored to try to transfer such a policy from simulation to the real world, such as domain randomization (Tobin et al., 2017) or building a more accurate simulator (Tan et al., 2018; Chebotar et al., 2018). However, obtaining an accurate simulator is often very challenging, especially if the simulator differs from the real-world in unknown ways. Further, building the simulator itself can be fairly complex. Because these methods require the ground-truth state to obtain the reward function, they require training in a simulator and do not allow further fine-tuning after deployment in the real world; our method, in contrast, does not require the ground-truth state for the reward function.

### 2.2 ROBOT LEARNING WITHOUT GROUND-TRUTH STATE

**Learning a reward function without supervision:** One line of work for learning a reward function is to first learn a latent representation and then derive the reward function based on the distance in the embedding space, such as cosine similarity. The representation can be learned by maximizing the mutual information between the achieved goal and the intended goal (Warde-farley et al., 2018), reconstruction of the observation with VAE (Nair et al., 2018), or learning to match keypoints with spatial autoencoders (Finn et al., 2016b). Recent work also explicitly learns a representation that is suitable for gradient based optimizer and then use it for specifying rewards (Yu et al., 2019b). However, there is no guarantee that these learned representation are suitable for deriving rewards for control. In addition, if the representation is pre-trained, it may also be incorrect in some parts of the observation space which can be exploited by the agent. Our approach is much simpler in that

the reward function does not have any parameters that need to be learned and we empirically show better performance to some reward learning approaches.

**Changing the optimization:** Another approach is to forego maximizing a sum of rewards as is typically done in reinforcement learning and instead optimize for another objective. For example, one method is to choose one-step greedy actions based on a learned one-step inverse dynamics model; after training, the policy is then applied directly to a multi-step goal (Agrawal et al., 2016). An alternative method is to learn a predictive forward dynamics model directly in a high-dimensional state space and use visual model-predictive control (Finn et al., 2016a; Finn & Levine, 2017; Ebert et al., 2017; 2018a;b). Although these methods have shown some promise, predicting future high-dimensional observations (such as images or depth measurements) is challenging. Another approach is to obtain expert demonstrations and define an objective as trying to imitate the expert (Sermanet et al., 2018; 2016; Peng et al., 2018; Finn et al., 2017). Our approach, however, applies even when demonstrations are not available.

### 2.3 Manipulating Deformable Objects

Deformable object manipulation presents many challenges for both perception and control. One approach to the perception problem is to perform non-rigid registration to a deformable model of the object being manipulated (Huang et al., 2015; Lee et al., 2015; Schulman et al., 2013; Wang et al., 2011; Javdani et al., 2011; Miller et al., 2011; Cusumano-Towner et al., 2011; Phillips-Grafflin & Berenson, 2014). However, such an approach is often slow, leading to slow policy learning, and can produce errors, leading to poor policy performance. Further, such an approach often requires a 3D deformable model of the object being manipulated, which may be difficult to obtain. Our approach applies directly to high-dimensional observations of the deformable object and does not require a prior model of the object being manipulated.

## 3 Problem Formulation

In reinforcement learning, an agent interacts with the environment over discrete time steps. In each time step $t$, the agent observes the current state $s_t$ and takes an action $a_t$. In the next time step, the agent transitions to a new state $s_{t+1}$ based on the transition dynamics $p(s_{t+1}|s_t, a_t)$ and receives a reward $r_{t+1} = r(s_t, a_t, s_{t+1})$. The objective for the agent is to learn a policy $\pi(a_t|s_t)$ that maximizes the expected future return $R = \mathbb{E}\left[\sum_{t=0}^{\infty} \gamma^t r_{t+1}\right]$, where $\gamma$ is a discount factor.

### 3.1 Goal-reaching Reinforcement Learning

In order for the agent to learn diverse and general skills, we define a goal reaching MDP (Schaul et al., 2015; Andrychowicz et al., 2017) as follows: In the beginning of each episode, a goal state $s_g$ is sampled from a goal distribution $\mathcal{G}$. We learn a goal conditioned policy $\pi(a_t|s_t, s_g)$ that tries to reach any goal state from the goal distribution. We use a goal conditioned reward function $r_t = r(s_{t+1}, s_g)$ and optimize for $\mathbb{E}_{s_g \sim \mathcal{G}}\left[\sum_{t=0}^{\infty} \gamma^t r_t\right]$. The transition dynamics $p(s_{t+1}|s_t, a_t)$ of the environment remain independent of the goal.

In many real-world scenarios, it is often difficult to construct a well-shaped reward function. Past work has shown that sparse rewards, combined with an appropriate learning algorithm, can achieve better performance than poorly-shaped dense rewards in goal-reaching environments (Andrychowicz et al., 2017). We thus define a sparse reward function that only makes the binary decision of whether the goal is reached or not. Specifically, let $S_+(s_g)$ be a subset of the state space such that any state in this set is determined to be sufficiently close to $s_g$ (in some unknown metric); in other words, if the environmental state is within $S_+(s_g)$, then the task of reaching $s_g$ can be considered to be achieved.[1] Naturally, we can assume that $s_g \in S_+(s_g)$. A binary reward function can then be defined as

$$r(s_{t+1}, s_g) = \begin{cases} R_+ & s_{t+1} \in S_+(s_g) \\ R_- & s_{t+1} \notin S_+(s_g), \end{cases} \tag{1}$$

where $R_+$ and $R_-$ are constants representing the rewards received for achieving the goal and failing to achieve the goal, respectively.

---

[1] $S_+$ is a function that maps from the state space to a subset of the space.

## 3.2 REWARDS FROM IMAGES

In many cases, the ground-truth state $s_t$ is unknown and we cannot directly use the true reward function defined in Equation 1. Instead, the agent observes high-dimensional observations $o_t$ from sensors, from which we must instead define a proxy reward function $\hat{r}(o_{t+1}, o_g)$. The question now becomes how to choose $\hat{r}$ to be optimal for reinforcement learning?

The most common approach in robotics is to perform state estimation. However, in many cases, the state estimator might be hard to obtain, such as for deformable object manipulation, e.g. laundry folding or food preparation. We therefore investigate whether an alternative reward function that does not depend on state estimation can be used. Specifically, let us consider a general reward function defined in observation space of the form

$$\hat{r}(o_{t+1}, o_g) = \begin{cases} R_+ & o_{t+1} \in \hat{O}_+(o_g) \\ R_- & o_{t+1} \notin \hat{O}_+(o_g), \end{cases} \tag{2}$$

where $o_g$ is a representation of the goal in observation space and $\hat{O}_+(o_g)$ is a subset of the observation space for which we will give positive rewards. A number of past approaches have proposed various methods for learning an observation-based reward function $\hat{r}(o_{t+1}, o_g)$ (Warde-farley et al., 2018; Nair et al., 2018; Florensa et al., 2019; Yu et al., 2019b). However, these approaches do not analyze the properties needed by such a reward function to enable optimal learning. Next we will investigate trade-offs between different choices of $\hat{O}_+(o_g)$ and how they will affect policy training time when trained with rewards of $\hat{r}(o_{t+1}, o_g)$.

## 4 REWARD MISCLASSIFICATIONS

We will now investigate how to design a good proxy reward function $\hat{r}(o_{t+1}, o_g)$, based on raw sensor observations, that we can use to train the policy; we desire for the policy trained with $\hat{r}(o_{t+1}, o_g)$ to optimize the original reward $r(s_{t+1}, s_g)$ based on the ground-truth state (which we do not have access to). Our first insight into choosing a good

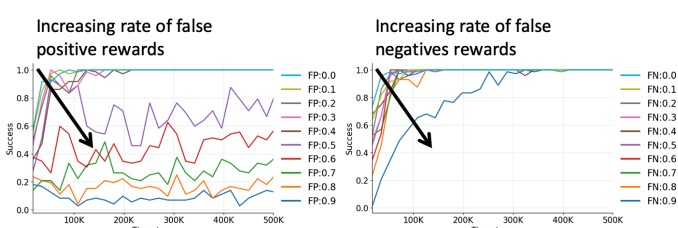

Figure 2: As we increase false negative/positive rewards, the learning curves with false positive rewards are affected more severely.

proxy reward function $\hat{r}(o_{t+1}, o_g)$ is that we should think about reward functions in terms of false positives and false negatives. Let us define a false positive reward to occur when the agent receives a positive reward based on our proxy reward function $\hat{r}(o_{t+1}, o_g)$ when it would have received a negative reward based on the original reward function $r(s_{t+1}, s_g)$. In other words, suppose that an unknown function $f$ maps from an observation to its corresponding ground-truth state, $s_t = f(o_t)$. The function $f$ exits as we assume that the environment is fully observable and no two states will have the same observation. Then a false positive reward occurs when $o_{t+1} \in \hat{O}_+(o_g)$ while $f(o_{t+1}) \notin S_+(s_g)$. Similarly, a false negative reward can be defined.

Intuitively, both false positive rewards and false negative rewards can negatively impact learning. However, for any estimated reward function $\hat{r}(o_{t+1}, o_g)$, we will have either false positives or false negatives (or both) unless we have access to a perfect state estimator. To design a good proxy reward function, we must ask: which will more negatively affect learning: false positives or false negatives?

The two types of mistakes are not symmetric. As we will see, a false positive reward can significantly hurt policy learning, while a false negative reward is much more tolerable. Under a false positive reward, the agent receives a positive reward (under the proxy reward function $\hat{r}(o, o_g)$) for reaching some observation $o$, even though the agent should receive a negative reward based on the corresponding ground-truth state $f(o)$ under the original reward function $r(s_t, s_g)$. This false positive reward will encourage the agent to continue to try to reach the state $f(o)$, even though reaching this state does not achieve the original task since $f(o) \notin S_+(s_g)$.

On the other hand, false negative rewards are much more tolerable. Under a false negative, the agent observes some observation $o$ such that $f(o) \in S_+(s_g)$ but the agent receives a negative reward.

However, if the agent still receives a positive reward for some other observation $o'$ such that $f(o') \in S_+(s_g)$, then the agent can still learn to reach the goal states $S_+(s_g)$, though learning might be slower and the learned policy may be suboptimal.

We provide a simple example to verify this intuition. Consider a robot arm reaching task, with the observation space $\mathcal{O} \in \mathcal{R}^3$ being the 3D position of the end-effector (EE). Note that here the observation space coincides with the state space and the agent directly observes the states. The action space $\mathcal{A} \in \mathcal{R}^3$ controls the position of EE. The true reward is defined by $S_+(s_g) = \{o \mid \|s_t - s_g\|_2 < \epsilon\}$. We define two types of noisy reward functions used for training. The reward function $\hat{r}_{FP}$ gives the same rewards as the true reward function, except that with a probability of $p_{FP}$ (False Positive Rate), a negative reward will be flipped to a positive reward. The reward function $\hat{r}_{FN}$ can be similarly defined, where a positive reward will be flipped to a negative reward with a probability of $p_{FN}$ (False Negative Rate). For this experiment, we use a standard reinforcement learning algorithm DDPG (Lillicrap et al., 2016) combined with goal relabeling (Andrychowicz et al., 2017). The learning performance of this same algorithm with different noisy rewards can be shown in Figure 2. We can see that the agent is able to learn the task even with a very large false negative rate. But when the false positive rate increases, the performance sharply decreases.

## 5 APPROACH

### 5.1 INDICATOR REWARDS

Following this idea, we propose using a proxy reward function that does not have any false positive rewards. To do so, we will use an extreme reward function of $\hat{O}_+(o_g) = \{o_g\}$. In other words, we will use an indicator reward function:

$$\hat{r}_{ind}(o_{t+1}, o_g) = \begin{cases} R_+ & o_{t+1} = o_g \\ R_- & o_{t+1} \neq o_g, \end{cases} \tag{3}$$

It should be clear that this reward function will have no false positives, since the reward is positive only if $o_{t+1} = o_g$, which implies that $f(o_{t+1}) = f(o_g)$, or equivalently, $s_{t+1} = s_g$. As $s_g \in S_+(s_g)$ by definition, all positive rewards are true positives. However, this reward function is extremely sparse and has many false negatives. In fact, without goal relabeling, in continuous state spaces, we would expect all rewards to be negative under this indicator reward function, as no two observations in continuous spaces will ever be identical. Next, we will describe how to learn with this reward function with goal relabeling.

### 5.2 GOAL RELABELING FOR OFF-POLICY LEARNING

Fortunately, for off-policy multi-goal learning, we can adopt the goal relabeling technique introduced in (Kaelbling, 1993; Andrychowicz et al., 2017) to learn the goal-conditioned Q-function. Suppose that some transitions $(o_t, a_t, o_{t+1})$ are observed when the agent takes an action $a_t \sim \pi(o_t, o_g)$ with a goal of $o_g$. Because Q-learning is an off-policy reinforcement learning algorithm, we can replace the goal observation $o_g$ with any other observation $o_{g'}$ in our Bellman update of the Q-function. This works because the transition dynamics $p(s_{t+1}|s_t, a_t)$, and likewise the observation transition dynamics $p(o_{t+1}|o_t, a_t)$, are independent of the goal $o_g$. Specifically, for some transitions, we will choose to replace $o_g$ with the observation $o_{t+1}$. By re-labeling $o_g$ with $o_{t+1}$ and using our indicator reward function, we will have that $\hat{r}_{ind}(o_{t+1}, o_g) = \hat{r}_{ind}(o_{t+1}, o_{t+1}) = R_+$. Thus, using goal relabeling, we can get positive rewards, even when using an indicator reward function in continuous state spaces.

### 5.3 REWARD BALANCING AND FILTERING

As mentioned above, after sampling a batch of data, we train the Q-function with goal relabeling. We use three different strategies for choosing which goals to use for relabeling: with probability $p_1$, we relabel $o_g$ with $o_{t+1}$, which will receive a positive reward under our indicator reward function. With probability $p_2$, we relabel the goal $o_g$ with $o_{t'}$ with an observation from some future time $t'$ step within the episode. The indicator reward function will most likely give a negative reward in this case, which is possibly a false negative because the new goal is possibly considered achieved based on the state-based, ground-truth reward funciton. Finally, with probability $p_3$, we use the original

goal (with no relabeling), which will again most likely give a negative reward under the indicator reward function; as before, this might be a false negative.

**Reward balancing:** We refer to "reward balancing" as setting $p_1 = p_2 = 0.45$ and $p_3 = 0.1$, leading us to receive positive rewards approximately $0.45$ of the time and negative rewards approximately $0.55$ of the time. Thus the ratio of positive and negative rewards that we use to train the Q-function are approximately balanced, even with indicator rewards. From another perspective, $p_1$ and $p_2$ determine the relative frequency between providing positive rewards and propagating rewards to other timesteps in the episode. Additionally, training with a small fraction of the original goals (i.e. $p_3$) can be seen as a regularization which ensures that the distribution of the relabeled goals moves towards the original goal distribution.

**Reward filtering:** While false negative rewards do not hurt learning as much as false positives, we still wish to avoid them if possible to improve the convergence time of the learned policy. We achieve this using "reward filtering," in which we filter out transitions that we suspect of having a high chance of being false negatives. We refer to "reward filtering" as discarding a sampled transition if its Q value is above a threshold $q_0$. If the assigned reward is negative based on the proxy reward function but the Q-value is sufficiently high, then there is a chance that this reward a false negative. To reduce the fraction of false negatives, we filter out such transitions and do not use them for training.

We can estimate how to set the threshold $q_0$ as follows: for a given transition $(o_t, a_t, o_{t+1})$, if $o_{t+1} = o_g$, we know that $\hat{r}_{ind}(o_{t+1}, o_g) = R_+$. In this case, $Q^*(o_t, a_t, o_g) = R_+/(1 - \gamma)$, where $Q^*$ is the optimal Q-function, assuming the optimal policy will continue to receive (discounted) positive rewards in the future. Similarly, if $o_{t+1} \neq o_g$, then $\hat{r}_{ind}(o_{t+1}, o_g) = R_-$. Since we know that the policy starting from $o_t$ will thus receive at least one negative reward before receiving positive rewards, then $Q^*(o_t, a_t, o_g) \leq R_- + \gamma R_+/(1 - \gamma)$. Thus, we can set a threshold $q_0$, where $R_- + \gamma R_+/(1 - \gamma) < q_0 < R_+/(1 - \gamma)$; if we find that $Q(o_t, o_g, a_t) > q_0$, then the corresponding reward $\hat{r}_{ind}(o_{t+1}, o_g)$ is likely to be a false negative (assuming that the Q-function has been trained well); we thus filter out such rewards, to reduce the number of false negatives that we use for training. We can see that $q_0$ is set to a rather conservative fitlering value. Additionally, the Q-function is initialized to a relatively low value to avoid overestimation of the Q-function which can lead to incorrect filtering in the beginning of the training.

# 6 ANALYSIS

In this section, we analyze the performance of learning with indicator rewards. We first interpret the goal conditioned Q-function as a measure of the time to reach one observation from another.

## 6.1 MINIMUM REACHING TIME INTERPRETATION

Let us define $d = D_\pi(o_t, a_t, \hat{O}_+(o_g))$ as the number of time steps it takes for the policy $\pi$ to go from the current observation $o_t$, starting with action $a_t$, to reach the set $\hat{O}_+(o_g)$ of goal observations. For simplicity, we assume that, once the agent receives a positive reward, it will take actions to continue to receive positive rewards. The Q-function can be written as

$$Q_\pi(o_t, a_t, o_g) = R_- + \gamma R_- + ... + \gamma^{d-1} R_- + \gamma^d R_+ + \gamma^{d+1} R_+ + ...$$
$$= \frac{\gamma^d}{1 - \gamma}(R_+ - R_-) + \frac{1}{1 - \gamma} R_-. \tag{4}$$

Now it can be easily seen that, as long as $R_+ > R_-$, $Q_\pi$ is strictly monotonically decreasing w.r.t. $d$. As such, maximizing $Q_\pi$ over $\pi$ is equivalent to minimizing the time the agent takes to reach the goal $\hat{O}_+(o_g)$. Note that this is true for varying definitions of $\hat{O}_+(o_g)$; thus the policy trained under the true reward function (Equation 1) will minimize $D_\pi(o_t, a_t, O_+(o_g))$ whereas the policy trained under the indicator reward function will minimize $D_\pi(o_t, a_t, \{o_g\})$ (slightly overloading notation for $D_\pi$). Below we will show how this interpretation of the policy's behavior at convergence can lead to a simple analysis of the suboptimality of the learned policy under the indicator reward.

## 6.2 ANALYSIS OF SUB-OPTIMALITY

Due to the false negative rewards given by the indicator function $\hat{r}_{ind}$, the learned policy may not be optimal with respect to the original reward function $r(s_{t+1}, s_g)$ defined in Equation 1. Here we

give the worst case bound for the policy learned with the indicator reward. Following the minimum reaching time interpretation of the previous section, we evaluate the performance of the policy in terms of the time it takes to reach the set of goal observations $O_+(o_g)$ from the current observation. Given $o_t, o_g$, denote $t_1$ as the minimum number of time steps to reach from $o_t$ to the set of true goal observations, i.e. $t_1 = D(o_t, O_+(o_g))$. Let $t_2 = D(o_t, o_g)$ be the minimum time to reach from $o_t$ to $o_g$. Define the diameter of this goal observation set as $d = max\{D(o_1, o_2)|o_1, o_2 \in O_+(o_g)\}$. From the optimality of $t_2$, we know that

$$t_2 \leq D(o_t, o) + D(o, o_g), \forall o \in O_+(o_g).$$

Thus,

$$t_2 \leq \min_{o \in O_+(o_g)} D(o_t, o) + D(o, o_g) \leq t_1 + d. \tag{5}$$

From the analysis in the previous section, the optimal policy which optimizes the indicator reward will reaches $o_g$ in $t_2$ time steps; since $o_g \in O_+(o_g)$, we know that this policy will reach $O_+(o_g)$ in some time $t_3 \leq t_2$. Also recall that we have defined $t_1$ such that the optimal policy under the true reward function of Equation 1 will reach $O_+(o_g)$ in $t_1$ steps. Thus $t_3/t_1 \leq (t_1 + d)/t_1$ is an upper bound on the suboptimality of the policy trained under the indicator reward, at convergence.

## 7 EXPERIMENTS

Our experiments address the following questions:

1. In the case of visual input, how much are the sample efficiency and the final performance affected without assuming access to the ground-truth reward?
2. How much does reward balancing and filtering improve learning efficiency?
3. Can our method scale to real world robotic tasks?

We denote our method, which uses indicator rewards with reward balancing and filtering, as **Indicator+Balance+Filter**. We compare our method with the following baselines:

- **Oracle**: This method assumes access to the ground-truth reward from state space $r(s_t, s_g)$.
- **Indicator**: This is an ablation of our method, without reward balancing and filtering.
- **Auto Encoder (AE)**: We train an autoencoder with an $L2$ reconstruction loss of the image observation, jointly with the RL agent. We then use cosine similarity in the learned embedding space to provide dense rewards, as similarly compared in (Warde-farley et al., 2018). Specifically, assuming the learned encoding of an observation $o$ is $\phi(o)$ after $L2$ normalization, the reward will be $r(o, o_g) = max(0, \phi(o)^T \phi(o_g))$.
- **Variational Auto Encoder (VAE)**: Similarly to the AE baseline, a VAE is jointly trained with the RL agent to provide rewards, as done in (Nair et al., 2018). For a fair comparison, the goal sampling strategy for this baseline is kept the same a other approaches.
- **Distributional Planning Network (DPN) (Yu et al., 2019b)**: DPN aims to learn a representaiton that is suitable for a gradient based planner to reach a goal observation. Following (Yu et al., 2019b), we first pre-train DPN using randomly generated samples and then use the learned representation for giving rewards.

Only Oracle uses the ground-truth, state-based reward function. We use the standard off-policy learning algorithm DDPG (Lillicrap et al., 2016) with goal relabeling (Andrychowicz et al., 2017). For methods without reward balancing, we re-label the current goal with an achieved goal sampled uniformly from one of the future time steps within the same episode with a probability of 0.9; otherwise, the original goals are used. For all the environments, the ground-truth rewards are based on the $L_2$ distance in the state space: $R_+$ if $\|s_{t+1} - s_g\| \leq \epsilon$ and $R_-$ otherwise. More details on algorithms, architectures, and hyperparameters can be found in the Appendix.

We first evaluate all the methods in a set of simulated environments in MuJoCo (Todorov et al., 2012), where both current and goal observation are given by RGB-D images:

- Reacher: Teach a two-link arm to reach a randomly located position in 2D space.

- FetchReach: Move the end effector of the Fetch robot to a random position in 3D.

- RopePush: Push a rope from a random initial configuration into a target configuration.

The first two environments above are standard environments from Gym (Andrychowicz et al., 2018a). For the more complex RopePush task, the robot needs to push a 15-link rope to a targeted pose, as shown in Figure 1. To accelearate learning, we fix the orientation of the gripper and parameterize the action as $(x_1, y_1, x_2, y_2) \in \mathcal{R}^4$, denoting the starting and ending position of one push from the gripper. We generate the initial rope pose by giving the rope a random push from a fixed location. The goal poses are generated by giving the rope two more pushes based on the initial push (these pushes are hidden from the policy). The robot can give three pushes to push the rope to the goal pose. More details on environments can be found in Appendix A.

The results are shown in Figure 3. We see that our method (Indicator+balance+Filter) achieves nearly the same performance as the Oracle even though our method operates only from RGB-D images and does not have access to the ground-truth state. For the RopePush environment, only the Oracle and our method are able to learn to achieve the task to a reasonable accuracy. For AE, VAE and DPN, the learned representation may not lead to a perfect reward function everywhere and the agent will exploit states that yield a high proxy rewards, even though the goal is not achieved. Instead, our method does not require learning a reward function and outperforms these baselines.

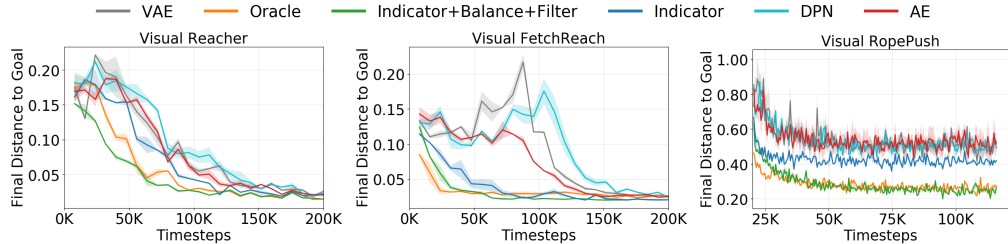

Figure 3: The final distance to goal of different methods in different environments throughout the training. The observations are from the RGB-D images rendered in simulation.

In Appendix C, we perform an ablation where we removed either Balance or Filter from our method. These experiments show that both Balance and Filter are important for optimal performance across the environments tested. In Appendix D, we show experiments in which the ground-truth states are used as inputs to the policy but are not used to compute the rewards. In these experiments, we also see that our method has a similar performance to the Oracle and outperforms the other approaches.

## 7.1 INDICATOR REWARDS WITH REAL IMAGES

Using RGB-D observations and goals, we train a Sawyer robot for a 3-dimensional reaching task. Figure 4 shows example observation and goal images as well as the distance to goal throughout training. As before, our method (Indicator+Balance+Filter) performs similarly to the Oracle in terms of final goal distance; the baseline of Indicator Rewards without balancing or filtering performs significantly worse and diverges in the end due to many negative rewards, many of which are false negative.

Figure 4: An example observation image (top left) and goal image (top right); final distance to goal (bottom).

## 8 CONCLUSION

In this work, we show that we can train a robot to perform complex manipulation tasks directly from high-dimensional images, without requiring access to the ground-truth state in either the policy input or the reward function. We empirically show that our method enables a robot to learn complex skills for manipulating deformable objects, for which state estimation is often challenging, including a real-world experiment.

We provide a theoretical analysis which shows that the optimal policy under the indicator reward has a bounded sub-optimality compared to the optimal policy under the ground-truth reward. We hope that our method will enable robot learning in the real world in cases where it is difficult to add extra sensors or accurately simulate the environment.

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

## A  ENVIRONMENT DETAILS

During evaluation, for all environments, a binary sparse reward is given at each time step. A positive reward $R_+ = 1$ is given when the goal is reached, i.e. $||s_{t+1} - s_g|| \leq \epsilon$ and a negative reward $R_- = -1$ is given otherwise. Other environment details are summarized in Table 1.

For the RopePush environment, to save the time for computing initial and goal configuration, 10,000 initial configurations of the rope are pre-computed and cached which are later used for training.

| Environment | Observation Dimension | Goal Dimension | Rendered Dimension | Action Dimension | Horizon (T) | $\epsilon$ (m) |
|---|---|---|---|---|---|---|
| Reacher | 10 | 2 | 100x100x3 | 2 | 50 | 0.01 |
| FetchReach | 10 | 3 | 100x100x4 | 3 | 50 | 0.05 |
| FetchPush | 25 | 3 | - | 4 | 50 | 0.05 |
| RopePush | 45 | 30 | 100x100x3 | 4 | 3 | 0.1 |
| VisualReacher (Sawyer) | - | - | 100x100x4 | 3 | 25 | 0.1 |

Table 1: Summarized environment details. The observation and goal dimension are the dimensions of the low dimension state representation when available. The rendered dimensions are the dimensions of the rendered RGBD images used in the visual experiments.

**Sawyer robot experiment details**: The observation is recorded with an Intel RealSense D435 depth camera. The goal observations are sampled by moving the robot arm to a uniformly sampled location in a cuboid of diagonal length 1.3m. The episode was considered successful if the end effector moved to within 0.1 m from goal location at the end of the episode (we used a time horizon of 25 steps). The trained policy performs position control and outputs end effector displacement within a range of -0.05m to 0.05m in each direction.

## B  HYPER-PARAMETERS

All the experiments are run for two random seeds. The hyper-parameters of the training algorithm with indicator rewards are summarized in Table 2. For all experiments with visual observation, the parameters of the convolution layers are shared among the observation input and goal input. Due to the complexity of the RopePush environment, a spatial softmax layer (Levine et al., 2016) with an output size of 32 is applied before the fully connected layers.

| Parameter | Value |
|---|---|
| *shared* | |
| positive reward ($R_+$) | 1 |
| negative reward ($R_-$) | -1 |
| reward filtering ($q_0$) | $\frac{1}{2}\left[R_- + \gamma R_+/(1-\gamma) + R_+/(1-\gamma)\right]$ |
| optimizer | Adam (Kingma & Ba, 2014) |
| learning rate | 0.001 |
| discount ($\gamma$) | $\frac{T-1}{T}$ |
| target network smoothing ($\tau$) | 0.98 |
| nonlinearity | tanh |
| *state observation* | |
| replay buffer size | $10^6$ |
| minibatch size | 256 |
| network architecture | 3 hidden layers with 256 neurons for each |
| *visual observation* | |
| replay buffer size | $5 \cdot 10^3$ |
| minibatch size | 128 |
| network architecture | 4 convolution layers followed by 3 hidden layers with 256 neurons for each |

Table 2: Summarized hyper-parameters.

## C  ABLATION ANALYSIS

We show different ablations of our methods in Figure 5. We can see that, in the RopePush environment, filtering is required for the policy to learn. On the other hand, the Reacher and FetchReach environments show that balancing is required for optimal performance. In all cases, Indicator+Balance+Filter consistently performs better than all the ablated methods. Thus, these results show that both balancing and filtering are important for optimal performance across a range of tasks.

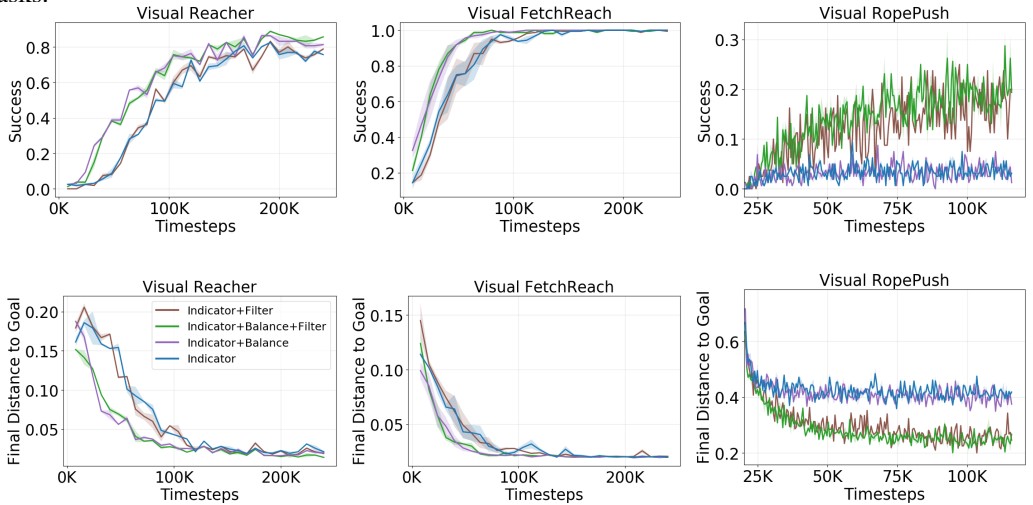

Figure 5: The success (top) and the final distance to goal (bottom) of different ablations of our method.

With indicator rewards, we do not have any false positive rewards but may have many false negative rewards. In Figure 6, we show the accuracy of the given rewards using different approaches with indicator rewards. The ground-truth rewards are calculated based on the ground-truth states which we do not have during training and are only used for analysis. We can see that with a longer time horizon, reward balancing is more important for the Visual Reacher and Visual FetchReach environment, which significantly lower the false negative rates. On the other hand, both reward filtering and balancing are important in the Visual RopePush environment. In all the cases, the accuracy of the rewards are improved a lot and sometimes almost perfect with reward balancing and filtering and we can see that this is necessary for the good performance of using indicator rewards.

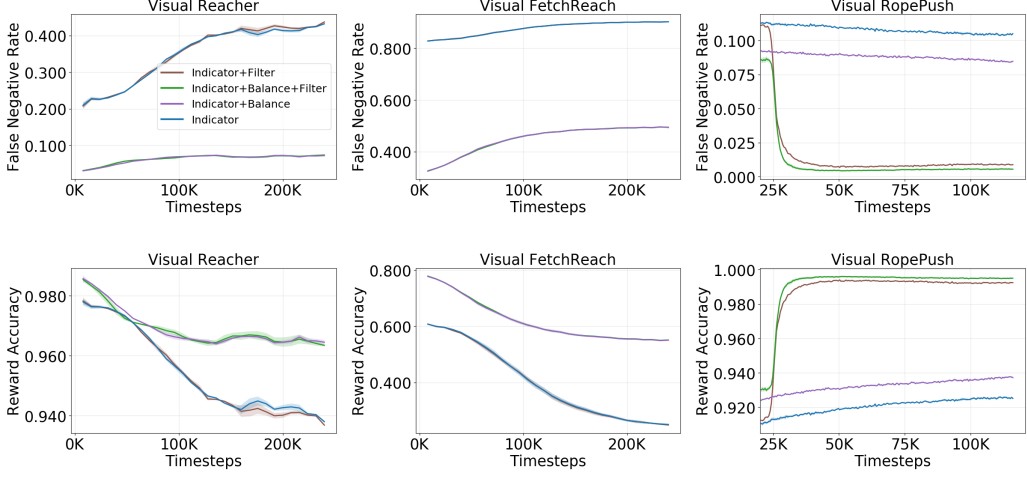

Figure 6: The false negative rate and the reward accuracy calculated from a batch sampled from the replay buffer of different ablations of our method.

# D    LEARNING WITH INDICATOR REWARDS WITH STATE INPUT

The performances of different learning methods when learning with low dimensional state representation are shown in Figure 7. In all the environments, using indicator rewards with reward balancing and filtering achieves comparable performances to Oracle. Compare this method to the Indicator baseline which does not have reward balancing and filtering, Indicator+Balance+Filter achieves a better sample efficiency. Interestingly, in the FetchReach environment, the default distance threshold for receiving an $R_+$ reward is set to 0.05. Thus, the policy that learns with this reward stops learning when the policy reach to such a distance to the goal, while the policy learned with indicator rewards keep reaching closer to the goals. This shows another benefit of using the indicator reward that the user does not need to tune the hyper-parameter $\epsilon$ to achieve the best performance.

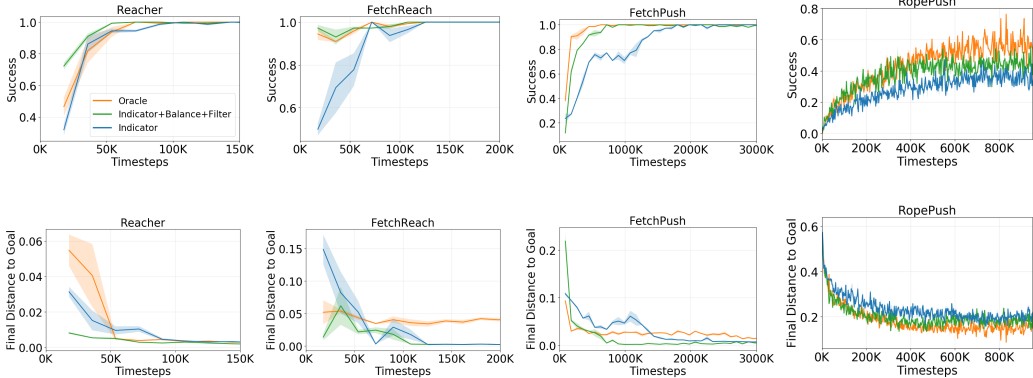

Figure 7: The success (upper row) and the final distance to goal (lower row) of different methods in different environments throughout the training. The success is defined as the mean probably of getting the $R_+$ reward over all time steps. The final distance to goal is defined as the L2 distance to the goal in the state space, in the last time step of the episode. The input to the policy is the low dimension state representation.

# E    IMPLEMENTATION DETAILS OF BASELINES

- **Auto Encoder (AE)**: The network architecture for the encoder is four convolutional layers each followed by a max-pooling layer. The latent code as a dimension of 32. The decoder is four up-pooling and up-convolution layers. The auto encoder is trained jointly with the RL agent by the reconstruction loss of the observation images in the sampled transition. The learning rate is set to 0.001. We then use cosine similarity in the learned embedding space to provide dense rewards, as similarly compared in (Warde-farley et al., 2018). Specifically, assuming the learned encoding of an observation $o$ is $\phi(o)$ after $L2$ normalization, the reward will be $r(o, o_g) = max(0, \phi(o)^T \phi(o_g))$.

- **Variational Auto Encoder (VAE)** The VAE uses the same architecture and training procedure as the AE baseline. Following (Nair et al., 2018), we use a variant of the VAE, $\beta$-VAE with $\beta = 5$. Following (Nair et al., 2018), given the latent representation $z, z_g$ of the current observation $o$ and the goal observation $o_g$, the rewards are given by

$$r(o, o_g) = -||z - z_g||_2,$$

which is the negative of the Euclidean distance.

- **Distributional Planning Network (DPN)** (Yu et al., 2019b) We use the released implementation (Yu et al., 2019a) and use the default hyperparameters. The network architecture for encoder is three convolutional layers, each with kernel size 5, stride 2. After each convolutional layer, there is layer normalization and sigmoid non-linearity. The output representation from CNN goes into a fully connected layer with latent code dimension of 128. For each environment, we collected 20,000 transitions from a random policy and performed 100,000 minibatch updates. The input dimension for each of the three environments is the same as the Rendered Dimension column specified in Table 1. Once the DPN is pre-trained, the representation is fixed during the training of the RL agent. Following (Yu et al., 2019b), given the latent representation $z, z_g$ of the current observation and the goal observation, the

rewards are given by

$$r(o, o_g) = -\exp(||d_{\text{DPN}}(z - z_g, \delta)||_1),$$

where

$$d_{\text{DPN}}(x, \delta)_i = \begin{cases} \frac{1}{2}x_i^2 & \text{for } |x_i| \leq \delta \\ \delta|x_i| - \frac{1}{2}\delta^2 & otherwise \end{cases}.$$

$\delta$ is set to 0.85. To avoid overflow of the rewards during the exponential, we normalize $z$ and $z_g$ such that they have a norm of 1 after the representation is trained.

## F  SENSITIVITY ON FILTERING THRESHOLD

In Section 5.3, we give the range of the filtering threshold $q_0$. In previous experiments, we use the middle value of this range. Here, we train with different values of $q_0$ within this range. The results for the rope pushing task is shown in Figure 8. We can see that the learning is very stable with different values of $q_0$.

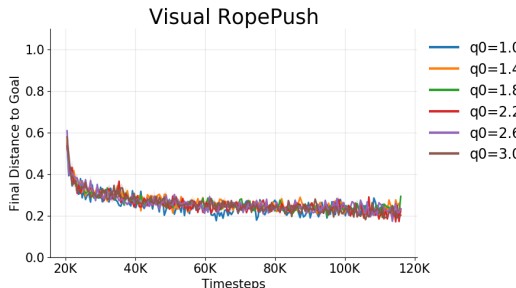

Figure 8: Learning with indicator rewards on the rope push environment using different filtering threshold. The range given in Section 5.3 is $[1, 3]$. Here we show 6 values of $q_0$ evenly spaced within this range.

