# OpenReview forum: "Reinforcement Learning without Ground-Truth State"
_ICLR.cc/2020/Conference — Reject_

### Official Review · AnonReviewer1 · 2019-10-21
**Official Blind Review #1**

**Rating:** 3

**Review:**

The authors propose to apply HER to image-based domain, assigning rewards based only on exact equality (and not using an epsilon-ball). The authors also propose to (1) filter transitions that are likely to cause false negative rewards and (2) balance the goal relabeling so that the number of positive and negative rewards are equal. The authors demonstrate that these two additions result in faster and better learning on a simulated 2d and 3d reaching, as well as a rope task. The authors also show that the method works on training a real-world robot to reach different positions from images.

Overall, the paper is a fairly straightforward and simple extension of HER, and the proposed changes seem to result in consistent improvements. Studying the importance of false positive vs false negative rewards is an interesting problem, but some of the details of the analysis are missing. Most of the writing of the paper is relatively clear, but there are a couple of assumptions that should be discussed more clearly. The analysis section seemed particularly confusing, as it introduces new notation and concepts (D without a subscript, goal observation set diameter) without explaining their significance. The experiments are relatively simple, making it difficult to judge whether or not the method will work on more challenging domains.

In more details:
1. The related works section feels unnecessarily long.

2. The analysis section is quite confusing to me and I do not understand its significance. It seems to say that reaching og rather than O+(og) is not very bad, if O+(og) is small. While this makes sense, it's not clear how the derivations of the paper add to this intuition. In particular, the paper derives an upper bound for t3/t1, which equals

 time to reach og / time to reach O+(Og)

Why is this "an upper bound on the suboptimality of the policy"?

3. Can details of Figure 2 be given? This seems like a very important experiment, so it is a shame that it is not discussed or analyzed in detail.

4. An assumption that is critical to this method's applicability is the assumption that no two states will receive the same observation. Without this assumption, the statement, "the reward is positive only if ot+1 = og, which implies that f(ot+1) = f(og), or equivalently, st+1 = sg" is false, and so therefore the statement, "It should be clear that this reward function will have no false positives" is also false. I do think that this is a reasonable assumption to make for some domains, but the authors should make that explicit.

5. For the "Indicator" baseline, what is the probability used for relabeling with a future state? I assume p1=0, but what are p2 and p3?

6. The ablations (Section C, Figure 5) indicates that only the filtering matters. I don't think this detracts from the author's submission, but it would be good for the authors to highlight the main paper. Currently, the statements, "These experiments show that **both** Balance and Filter are important for optimal performance across the environments tested." seems unsubstantiated by the experiments--only filtering seems important.

7. The authors should clarify the difference between the "true reward" and the "original reward." What is the relationship? The last paragraph of Section 4 discusses false positive rates and says "true reward is defined by O+(og) = {o | ||o − og||2 < epsilon}." However, two paragraphs earlier, the paragraph also discuses false positive rewards and says, "under the original reward function r(st, sg)." The experiment section then says "Oracle uses the ground-truth, state-based reward function." Is the assumption that the "true" reward (defined in image space) is a perfect representation for the "original" reward (defined in state space)? Does this method work if this assumption is violated?

8. (very important) How sensitive is the method to q0, and how was it chosen for the experiments?

9. Figure 6 in the Appendix is very difficult to read.

I would be inclined to raise my score if the authors
a. clarified some questions above
b. conducted ablations to study the importance of q0, the threshold
c. added experiments on more challenging domains (e.g. cube pushing, pickup) that demonstrated the effectiveness of the method on more domains

**Experience Assessment:**

I have published one or two papers in this area.

**Review Assessment: Checking Correctness Of Derivations And Theory:**

I carefully checked the derivations and theory.

**Review Assessment: Checking Correctness Of Experiments:**

I carefully checked the experiments.

**Review Assessment: Thoroughness In Paper Reading:**

I read the paper thoroughly.

---

> ### Author Response · Authors · 2019-11-12
> **Response from the authors**
>
> Thank you for your feedback. Here are our responses to clarify some of your questions: (The numbering refers to the corresponding point in the review)
>
> 2. With indicator rewards, our policy aims to learn a policy that is optimal among all policy that reaches Og, while the optimal policy with respect to the ground truth rewards is optimal among all policy that reaches O_+(Og). Thus, this derivation would show that our policy, while sub-optimal, will not be too bad to an optimal policy that uses the ground truth rewards.
>
> 3. The details are provided at the last paragraph of Section 4
>
> 5. For the indicator baseline, for every goal to be relabeled, we uniformly sample from an achieved state in the future of this episode. Thus, p2 and p3 are state dependent.
>
> 6. Reward filtering is motivated in a principled way as shown in Section 5.3 and can be understood from the minimum reaching distance interpretation of the Q function. As such, in the last paragraph of Section 5.3, we provide the upper and lower bounds of the filtering hyperparameter q0. As explained in Appendix B, we simply choose the average of the upper and lower bound for all our experiments.
>
> We performed an additional experiment to evaluate how robust our method is to this specific choice of q0. For the rope pushing experiment, we evaluate multiple values of q0, evenly spaced within this lower and upper bound. We find that each choice of q0 in this range achieves the same learning speed and final performance as using the average of the bounds; this demonstrates that our method is not overly sensitive to the particular choice of q0, as long as it lies within the range of the lower and upper bounds described in Section 5.3.

---

> > ### Comment · AnonReviewer1 · 2019-11-15
> > **Re: Response from the authors**
> >
> > 2. I see. It's still unclear though why this adds to the intuition that, "reaching og rather than O+(og) is not very bad, if O+(og) is small."
> > 3. I see. Was this using the same reaching environment as in the HER paper?
> > 4. Not addressed.
> > 5. What are the values of p2 and p3? Are they the standard values from HER?
> > 6. I understand the intuition, but I don't think my point was addressed: Currently, the statements, "These experiments show that **both** Balance and Filter are important for optimal performance across the environments tested." seems unsubstantiated by the experiments--only filtering, empirically, seems important on the visual tasks.
> > 7. Not addressed.
> > 8. Please update the paper with the experimental results.

---

> > > ### Author Response · Authors · 2019-11-15
> > > **Response to R1**
> > >
> > > 2. Assuming the policy learned with indicator rewards is able to reach Og, then as O_g \in O+(Og), the policy also finishes the task. The learned policy is worse than the optimal policy, as it takes longer for the learned policy to reach O+(Og) but the additional distance it traverses is bounded by d.
> > > 3. This is the same reaching environment in HER, but with visual observation. We also test the algorithms with state input, shown in Appendix D
> > > 4. We have added this assumption in the first paragraph of section 4 in the updated paper.
> > > 5. In Section 5.3, we describes that p1=p2=0.45 and p3=0.1 and we refer to this as reward balancing. HER does not have the two values here.
> > > 6. In Figure 5 of Appendix C, you can see from the ablation experiments that in both Visual Reacher and Visual FetchReach environments, removing the reward balancing decreases the performance.
> > > 7. In the last paragraph of Section 4, in this example, we try to show the different effects of false positive and false negative rewards. In order to directly control the ratio of false positive and false negative rewards, in this example here, instead of having images as the observation, we directly give agents the states as observation, and add noise to the rewards to artificially create false positive and false negative rewards. We have clarified this example in the updated paper. Other than this example, we are consistent with the definition of the true rewards, which is the state based rewards that the agent needs to optimize and there are no additional assumptions here.
> > > 8. Please see Appendix E of the updated paper.

---

### Official Review · AnonReviewer2 · 2019-10-21
**Official Blind Review #2**

**Rating:** 1

**Review:**

The paper tackles the problem of self-supervised reinforcement learning through the lens of goal-conditioned RL, which is in line with recent work (Nair et al, Wade-Farley et al, Florensa et al, Yu et al.). The proposed approach is a simple one - it uses the relabeling trick from (Kaelbling, 1993; Andrychowicz et al., 2017) to assign binary rewards to the collected trajectories. They apply two simple tricks on top of relabeling:

1. Reward balancing: Balancing the number of 0/1 rewards used for training the policy.
2. Reward filtering: A heuristic that rejects certain negative-reward transitions for learning if the q value for the transition is greater than some threshold q_0.

While I like the simplicity of the proposed approach when compared to competing methods, my overall recommendation is reject based on the current state of the paper, because of the following reasons:

1. The technical novelty is quite limited - the paper mostly uses the framework from Andrychowicz et al., 2017 with a specific choice of epsilon (=0) for giving positive rewards. The method does reward balancing, but similar goal-sampling have been used in prior work like Nair et al., 2018, and is not essential to obtaining good results (Appendix C). The main technically novel component is likely the reward filtering mechanism, but I find it to be somewhat ad-hoc since it assumes that the Q-values learned by the Q-network to be reasonably good during training time, which is not the case for most modern Q-learning based methods [1, 2].
2. The provided analysis is not particularly illuminating, see my detailed notes below.
3. The experiments are underwhelming, see my detailed notes below.

I would be willing to overlook 1 or 2 if the authors did a more thorough experimental evaluation which showed the method working well when compared to alternatives, but that is not the case right now.

Note on Section 6 (analysis)
The authors provide a simple analysis in Section 6 to bound the suboptimality of the learned policy. Unless I’m missing something, the resulting bound of t_3 <= t_1 + d is trivially true, since d is defined to be the diameter of O_{+}(o_g), and t_1 is the number of timesteps taken by an optimal policy to go from o_t to O_{+}(o_g). As a result, I don’t find this analysis illuminating or interesting - perhaps the author can provide counter arguments here to change my mind.

Note on Section 7 (experiments)
For sim experiments, two of the tasks are extremely simple (free space reaching in 2D and 3D, respectively) where essentially everything works - the proposed method, baselines and ablations. The third task of rope manipulation is fairly interesting at a first look - but it appears to have been greatly simplified. The authors consider a high-level action space and an episode of only three timesteps. Further, the authors make the simulated robot arm invisible in the supplementary videos, which greatly simplifies the problem visually. Since the entire motivation is about learning well in real world settings, I feel this is a bit underwhelming. Figure 1 is misleading, since it shows a visible robot arm in front of a rope. This also appears to hint that the method did not work well with realistic visuals, highlighting a major limitation of the proposed approach. I think it would be valuable to include such failures (and discussions around them) in future submissions.

For the real world experiments, the task being considered is extremely simple (free space reaching), and does not even require pixel observations. Even for this simple task, the error achieved by the method is 10cm (starting error was 20cm), which is quite poor - robotic arms like the Sawyer should be able to achieve much lower errors. Even the oracle reward achieves an error of 10cm, which might indicate a bug in the author’s real world robotic setup. In comparison, prior work such as Nair et al. is able to tackle harder problems in the real world (like non-prehensile pushing).

Minor points
- Section 4 contains a nice discussion on false positives and false negatives when using non-oracle reward functions for reinforcement learning, where they also perform a simple experiment to show how false positives can negatively impact learning much more severely than false negatives. This does a good job of motivating the method (i.e. avoiding false positives), but also undermines the motivation behind reward filtering, which is perhaps the main technically novel component of the proposed approach.
- Section 2.3 (i.e. related work on deformable object manipulation) states that "Our approach applies directly to high-dimensional observations of the deformable object and does not require a prior model of the object being manipulated.”, and only cites prior work that assumes access to deformable object models. However, there is recent work that enable similar manipulation skills without access to such models. For example, Singh et al. [3] are able to learn to manipulate a deformable object (i.e. a piece of cloth) directly from high-dimensional observations using deep RL in the real world, and do not require object models (or ground truth state), but do require other forms of sparse supervision.
- Typo: On page 7, “is kept the same a other approaches.” -> “is kept the same as other approaches.”

[1]: Diagnosing Bottlenecks in Deep Q-learning Algorithms. Fu et al., ICML 2019
[2]: Double Q-learning. V. Hasselt. NIPS 2010
[3]: End-to-End Robotic Reinforcement Learning without Reward Engineering. Singh et al., RSS 2019.


**Experience Assessment:**

I have published one or two papers in this area.

**Review Assessment: Checking Correctness Of Derivations And Theory:**

I carefully checked the derivations and theory.

**Review Assessment: Checking Correctness Of Experiments:**

I carefully checked the experiments.

**Review Assessment: Thoroughness In Paper Reading:**

I read the paper thoroughly.

---

### Official Review · AnonReviewer3 · 2019-10-23
**Official Blind Review #3**

**Rating:** 3

**Review:**

### Summary ###

In this paper, the authors focus on the problem of goal conditioned reinforcement learning. Specifically, the authors consider the setting where the agent only observes vision as input and the ground truth state is not observable by the agent. In this setting, it is hard to specify a reward function since the reward function has to compute rewards from images.

The authors first consider the setting of using a proxy reward function, and argue that false positive errors in the proxy reward function hurt the policy training significantly more than false negative errors. The authors demonstrate this empirically in a simple simulated robot arm reaching setting. Then the authors propose to use an indicator reward function that eliminate all false positive errors. The authors combine the indicator reward function with a mixture of goal relabeling schemes and a heuristic way of filtering out data with false negative rewards.

The authors evaluated the proposed method on 3 simulated robotic manipulation environments and one real robot environment. The results presented by the authors suggest that the proposed method performs better than baseline methods.


### Review ###

Overall I think this paper presents an interesting idea in learning goal conditioned policies from vision. The idea is very well presented and authors include many empirical evidence to support the proposed method. However I do find a number of shortcomings that need to be addressed.


Pro:
1. The idea for this paper is really well presented. The structure of the paper is well organized and  the authors include informative explanations and empirical evidence to support the crucial assumption that false positive rewards are worse than false negative rewards. The results for the main experiments are also easy to understand.

2. The paper includes a fairly comprehensive set of ablation studies for each part of the proposed method in the appendix. The ablation study clearly illustrated the effects of balancing different goal relabeling schemes and filtering transitions.


Con:

1. I’m not convinced about the magnitude of novelty in this paper. The indicator reward has already been used in HER[1], and the balancing of relabeling schemes seems like a direct extension of the various relabeling schemes proposed in HER. It seems to me that the only novelty of this paper comes from the filtering techniques for false negative rewards, which I do not think is enough for this venue.

2. The experiment results are not very strong for the proposed method. In two of the three simulated robotics environments, the proposed method performs similarly to the indicator + balance configuration, which in my opinion is only a slight variation of HER. Therefore, I do not find the claimed advantage of the proposed method to be convincing.


The idea in the paper is well presented and carefully investigated. However, I am still not convinced about the novelty of the proposed idea and the magnitude of performance improvement. Therefore, I would not recommend acceptance before these problems are addressed.



References
[1] Andrychowicz, Marcin, et al. "Hindsight experience replay." Advances in Neural Information Processing Systems. 2017.


**Experience Assessment:**

I have published in this field for several years.

**Review Assessment: Checking Correctness Of Derivations And Theory:**

I assessed the sensibility of the derivations and theory.

**Review Assessment: Checking Correctness Of Experiments:**

I carefully checked the experiments.

**Review Assessment: Thoroughness In Paper Reading:**

I read the paper thoroughly.

---

### Author Response · Authors · 2019-11-12
**General response from the authors**

We thank the reviewers for the detailed feedback. We believe that our method, motivated by reducing the false positive rewards during RL, is an important alternative to the previous approaches that provide rewards based on a learned representation and the proposed reward filtering is a principled approach to reduce false negative rewards. However, we acknowledge that our current experiments except visual RopePush are done in relatively simple settings even though our algorithm still shows superior performance despite its simplicity.

---

### Decision · Program_Chairs · 2019-12-19

**Decision:**

Reject

**Comment:**

This paper considers the problem of reinforcement learning with goal-conditioned agents where the agents do not have access to the ground truth state.  The paper builds on the ideas in hindsight experience replay (HER), a method that relabels past trajectories with a goal set in hindsight.  This hindsight mechanism enables indicator reward functions to be useful even with image inputs.  Two technical contributions are reward balancing (balancing positive and negative experience) and reward filtering (a heuristic for removing false negatives).  The method is tested on multiple tasks including a novel RopePush task in simulation.

The reviewers discussed strengths and limitations of the paper.  One strength was that the writing was clear for the reviewers. One limitation was the paper's novelty, as most of these ideas are already present in HER with the exception of reward filtering.  Another major concern was that the experiments were not sufficiently informative.  The simulation tasks did not adequately distinguish the proposed method from the baseline (in two of the three tasks) and the third task (RopePush) was simplified substantially (using invisible robot arms).  The real world task did not require the pixel observations.  The analysis of the method was also found to be somewhat limited by the reviewers, though this was partially addressed by the authors.

This paper is not yet ready for publication since the proposed method has insufficient supporting evidence.  A more thorough experiment could provide stronger evidence by showing a regime where the proposed method performs better than alternatives.